# Mini-TAS as a confirmatory mapping tool for remapping areas with uncertain filarial endemicity to exclude/ include for mass drug administration: A report from field validation in India

Barsa Baisalini Panda[1], Kaliannagounder Krishnamoorthy[2], Arundhuti Das[1], Hitesh Kumar Jain[1], Sujata Dixit[1], Manju Rahi[3], Nilam Somalkar[4], Shubhashisha Mohanty[5], Sanghamitra Pati[1], Manoranjan Ranjit[1]*, Madhusmita Bal[1]*

1 Regional Medical Research Centre, Bhubaneswar, Odisha, India, 2 ICMR-Vector Control Research Centre, Puducherry, India, 3 Indian Council of Medical Research, New Delhi, India, 4 Regional Office for Health & Family Welfare, Bhubaneswar, Odisha, India, 5 National Centre for Vector Borne Diseases Control, Bhubaneswar, Odisha, India

* balmadhusmita@gmail.com (MB); ranjit62@gmail.com (MR)

## Abstract

India has targeted elimination of lymphatic filariasis (LF) through mass drug administration (MDA) by 2027. Mapping of LF endemic areas is a priority for implementation of MDA. Current national LF remapping tool for unsurveyed/uncertain districts, have many limitations. The WHO has recommended a sensitive and rapid remapping protocol (Mini-TAS), that needs validation in Indian setting. Hence, in the present study a comparative assessment of these two protocols (national protocol vs Mini-TAS) was undertaken in two non-MDA districts of Odisha, with unknown filarial endemicity but reporting chronic cases. Purposive sampling was done in five top sites based on filarial case count as per the national protocol. Random 30 cluster survey was done by conducting school based Mini-TAS, Microfilariae (Mf) survey among adults (>10 years) in villages/wards with schools and Molecular Xenomonitoring (MX) of infection in vectors. Costing by activity and items of the surveys was acomplished using itemized cost menu. In Kalahandi, one of the five purposive sampling sites showed Mf prevalence above threshold (> 1%). But except Mini-TAS neither MX nor household Mf survey among adults could detect the infection above the threshold. While in Balangir, Mf prevalence in all purposive sampling sites,Mini-TAS, Mf prevalence among adult and MX were above the respective thresholds confirming endemicity of LF in the district. The per sample cost of purposive sampling for Mf was the lowest INR 41, followed by adult Mf sampling INR 93. Mini-TAS and MX were expensive with INR 659 and 812 respectively. The study demonstrates that though all the sampling methods could detect filarial infection above the threshold in high-risk areas, Mini-TAS could only detect infection in low-risk areas. Therefore, in the national programme Mini-TAS can be used as a decision-making tool to determine whether to exclude/ include a district having uncertain endemicity for MDA.

**Data Availability Statement:** All relevant data are within the paper.

**Funding:** 1.Grant Received By: MB, MRR, SP 2. Grant Number: No. 6/9-7(248)/2020-ECD-II 3. Indian Council of Medical Research 4.https://main. icmr.nic.in/ 5.The funders had no role in study design, data collection and analysis, decision to publish, or preparation of the manuscript.

**Competing interests:** The authors have declared that no competing interests exist.

## Introduction

Lymphatic filariasis (LF), a neglected tropical disease is prevalent in 72 tropical and sub-tropical countries with about 1.4 billion people at the risk of infection [1]. A recent estimate has shown that about 51.4 million people are infected [2]. The Global Program to Eliminate Lymphatic Filariasis (GPELF) was launched in 2000 to eliminate lymphatic filariasis as a public health problem through mass drug administration (MDA) and morbidity management to alleviate suffering [3]. India, being a signatory of the 50th World Health Assembly resolution (WHA 50.29), launched National Program to Eliminate Lymphatic Filariasis (NPELF) in 2004 and a new global target of elimination by 2030 was set subsequently [4].

Enormous efforts have been made during the past two decades to eliminate LF in India, covering all the 328 endemic districts, spread over 16 states and 4 union territories. These endemic districts were identified based on the historical data and subsequent remapping. As of now MDA has been stopped in 99 districts, after successful demonstration of interruption of transmission [5]. However, a number of districts within the LF endemic states and other states excluded from MDA as they were considered to be either non-endemic or with uncertain endemicity due to lack of data during the initiation of the LF elimination program in 2004. However, it is mandatory for the program to provide the information on the current status of Mf in non-MDA districts in the dossier for obtaining validation of LF elimination as a public health problem [1]. According to the National Centre for Vector Borne Diseases Control (NCVBDC) guideline, line listing of filarial cases in non-endemic/non-MDA districts followed by Mf survey in eight sites, recorded with clinical cases (purposive sampling) is undertaken to confirm the endemicity of LF. Basically, purposive sampling targets only high-risk areas but may not be appropriate for the area under survey. To address this concern of uncertain endemicity, a new LF confirmatory mapping tool namely Mini-TAS has been developed by global LF experts during the year 2014 and piloted in Ethiopia and Tanzania [6,7], where school children within 9–14 years of age were selected from each school of the district either by systematic or cluster sampling method to assess the prevalence of circulating filarial antigen using Filariasis Test Strip (FTS). The rationale behind the confirmatory mapping tool is similar to that of the LF transmission assessment survey (TAS), in that both employ cluster sampling of children in schools and use a 2% threshold for decision-making [7]. Mini-TAS has several advantages than WHO protocol as it restricts and target the particular age group (9-14yrs) which is more likely to be the indicative parameter of recent transmission and also expanded the sampling sites to show the representative of the whole district. Therefore, the present study is carried out to compare these two mapping protocols to identify more sensitive and cost-effective mapping method besides assessing the feasibility of alternative methods such as Molecular Xenomonitoring (MX) and Mf prevalence among adult population for agreement in the decision on whether the area does or does not require MDA.

## Methodology

### Study area

The study was conducted during 2021 to 2022 in two districts viz., Kalahandi and Balangir of Odisha, an eastern Indian state known to be endemic for filariasis (Fig 1). The sample collection started from 5th July 2021 to 20th December 2022. The study districts are among the three districts which were considered non-endemic and LF elimination program is already being conducted in 27 out of total 30 districts in the state.

Kalahandi (19o 8" N 82⁰ 32" E) district is situated in the southwestern part of the state. The total population of the district is 1,573,054 and 28.7% of them belong to tribal communities

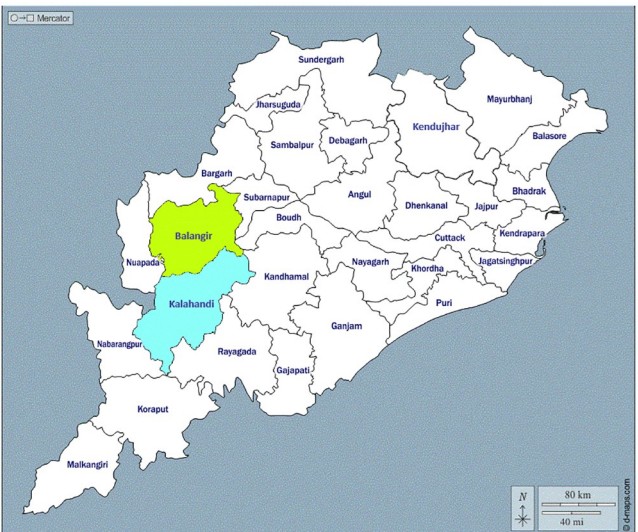

**Fig 1. Map of Odisha state showing the location of two study districts.** The map was downloaded from: https://d-maps.com/carte.php?num_car=272264&lang=en.

[8]. There are 4 towns (urban localities) and 2253 villages (rural settlements). About 92.3% of the people live in rural areas. Balangir (20˚7" N 83˚5" E) district is situated in the southern part of the state. The total population of the district is 1,648,997 and 21.1% of them belongs to tribal communities [8]. There are 8 towns (urban localities) and 1783 villages (rural settlements) in the district. Around 12% of people in the district live in urban areas while 88% live in rural areas.

## Sampling strategies and survey plans

Two different sampling strategies (clustered random sampling and purposive sampling) were adopted to compare the results in the two districts. Two epidemiological indicators (antigenaemia and Mf) and one entomological indicator (MX) were assessed for comparison and understand the risk of infection and transmission. The lowest (primary) administrative units such as villages in rural area and wards in urban area are used as primary sampling unit (PSU) and are referred as sites. A schematic representation of sampling strategy and survey plan is depicted in Fig 2.

**Purposive sampling.** The remapping protocol developed by the national program is to examine the data on morbidity survey through line listing of chronic cases by the trained health workers and conduct Mf survey in districts with chronic cases of LF. Mf survey is to be conducted in high risk sites (villages/wards) selected based on the number of cases. From the morbidity record of the district NCVBDC, list of villages reporting LF morbidity were obtained. These villages were sorted by number of cases and from the descending order of villages, first five sites were selected from this list for purposive sampling in each district. These villages were visited and sensitized the village head and community leaders. Household enumeration of family members were made by house to house visit and LF chronic cases were recorded by using pretested questionnaire. From each site, 500 individuals aged 5 years and above were screened for Mf, with household as the sampling unit. These individuals were from the households selected systematically using a sampling fraction, assuming a family size of target population of 5. The clinical cases were classified using the recommended WHO criteria.

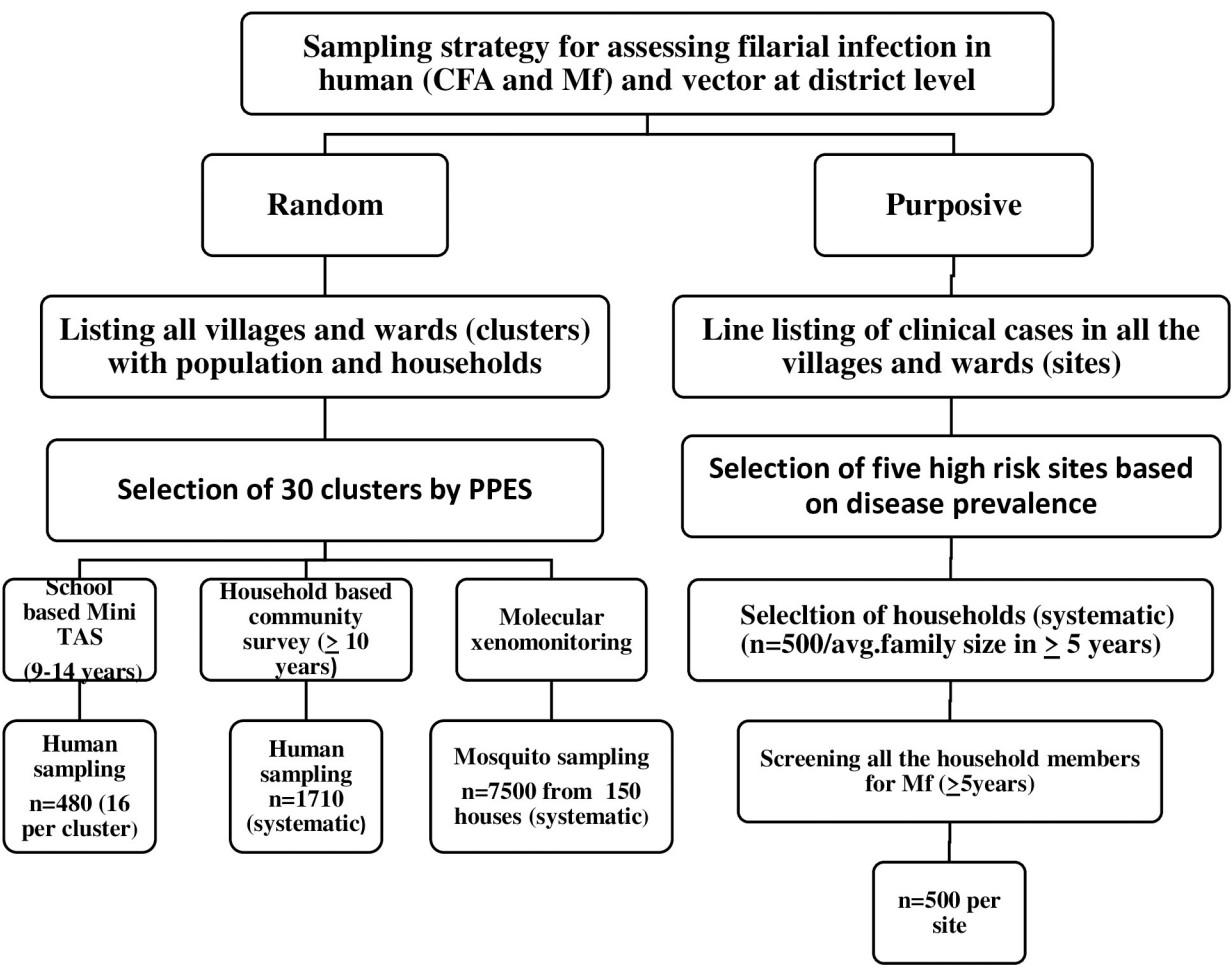

**Fig 2. Scheme of sampling strategies and survey plans in each study district.**

Migrated cases were excluded. Informed consent was obtained from the participating members of the family. Consent for children below 18 years was obtained from the parents/legal guardians. Night blood samples was collected between 19.30 hours and 23.30 hours and blood smears were prepared using 60 μl of finger prick blood.

**Random sampling.** A two-stage clustered random sampling was followed. In the first stage 30 sites were selected and in the second stage households were selected. The updated (2021) population and number of households for all villages and wards were obtained from community health centers, Municipalities and Notified Area Councils of the respective study districts and arranged as per their geographical proximity. Using Probability Proportionate to Size (PPS) method, 30 clusters were selected.

## Mini-TAS

While different methods [9] have been suggested for mapping LF, Lot quality assurance sampling(LQAS) based Mini-TAS was conducted in the schools located in the selected 30 sites following the Confirmatory Mapping Survey Sample Builder (SSB) tool [10] The sample size for Mini-TAS was 480 which is based on expected antigenemia prevalence 2% with 6% α error,

1.5 design effect and 40% power. All the schools in each of the selected site were included. In case of absence of school in the selected site, the nearest school was included and the children from the selected site alone were covered. A total of 16 children in grades 4th to 9th (age class 9–14 years) were systematically selected out of the total number of children in the target class/ age in each school. Where more than one school was present, all schools were included in the sample frame in the selected site, and desired samples were selected from the total children of the target class/age. Selected children were tested for circulating filarial antigen (CFA) using the FTS. The test line in the positive test was scored as 1–3 based on the line color intensity compared to control line. The evaluation area (here the district) is considered endemic when the number of CFA positive children is greater than the critical cutoff of 3 [6]. All the FTS positive children were tested for Mf using 60 μl finger prick night blood sample.

## Community Mf survey

Mf survey covering individuals above 10 years was conducted in the selected 30 random sites. The sample size for community survey was 1710 per district, estimated for a threshold of 1% Mf prevalence, with 5% chance of type 1 error, ~75% power and a design effect of 2.0 for populations >2800. Households in the selected random site were considered as the sampling unit (stage II). Assuming an average family size of 4 in the age class 10 years and above, the number of households to be selected was 430 which was distributed in all the selected 30 sites using a site-specific sampling fraction based on the population in each cluster. The number of houses to be visited and number of samples per site varied with the population size of the site. Finger-prick night blood sample (60 μl) was collected from the consenting individuals between 19.30 and 23.30 hours and prepared thick smear on a blood slide. Blood slides were dehemoglobinized in distilled water, dried, fixed in acidified methyl alcohol and stained using JSB-1 stain. Trained microscopists examined the blood slides and the number of Mf in each positive slide was recorded. All the positive slides and 5% of the negative slides selected randomly were cross-examined for quality assurance.

## Molecular Xenomonitoring

Adult female vector mosquitoes were collected from these 30 random sites for assessing vector infection. The sample size for Xenomonitoring was calculated to be 7500 female vector mosquitos (*Culex quinquefasciatus*) which was based on expected vector infection rate of 0.25%, with 80% power and 95% CI. With a target of two pools of 25 gravid mosquitoes per household, 150 households (stage II) were identified from 30 sites using systematic sampling from the total number of households in the sites. Gravid females were collected by using gravid traps (VCRC modified version of the CDC gravid trap, Model 1712, John W. Hock Co., Gainesville, FL) were placed outdoors within the household premises, at least 1 hour prior to sunset (1700 hours) after obtaining oral consent from the residents. Hay infusion was used as attractant for gravid *Cx. quinquefasciatu*s. The mosquito collection cages were removed from the traps the next morning (0600 hours) and brought to the field laboratory. Collected mosquitoes were morphologically identified to species and female *Cx.* q*uinquefasciatu*s were further classified according to their abdominal condition. Only gravid, semi-gravid and full-fed mosquitoes were used to make two pools of 25 female mosquitoes. With two pools from each selected house, a total of 300 pools were made. Traps were repeated for a maximum of three time to achieve the target in each house. Vials were barcoded and mosquitoes were dried at 95˚C for a minimum of 15 minutes using a dry bath. The pools were processed for assaying using PCR.

## Filarial DNA extraction from mosquito pools

Filarial parasite DNA was extracted from each mosquito pool using QIAGEN Insect Tissue extraction kit. The DNA samples thus obtained were coded and analyzed by multiplex PCR assay as described elsewhere [11]. Briefly, the PCR amplification was carried out in a final volume of 25 μl, which includes 10x buffer with 25mM magnesium chloride (MgCl2), 50mM MgCl2, 40mM of each dNTPs, 30 pmoles of forward universal primer (BM/WB F: 50- `AGC GTG ATG GCA TCA AAG TAG` -30) common to *B. malayi* and *W. bancrofti*, 20 pmoles of species specific reverse primers (BMR188 (*B malayi*): 50- `TTC CAT CCC CAA GAA AAT ATT AG` -30 and WBR129 (*W bancrofti*): 50- `AGG TTA TAC CAA GCA AAC AAA AA` -30) and 5 units of Taq DNA polymerase. PCR cyclic condition was initial denaturation for 5 min at 95˚C followed by 35 cycles of 1 min at 94˚C, 1 min at 50˚C, 2 min at 72˚C and a final extension of 10 min at 72˚C. The PCR assay was performed in a DNA Thermal cycler (Bio-Rad). After PCR amplification 15 μl of each amplified PCR product was electrophoresed on a 1.5% agarose gel containing ethidium bromide and compared with a 100-bp ladder (Merck) to confirm the amplicon size (*B. malayi*: 188 bp and *W. bancrofti*: 129 bp) under UV illumination and photographed in gel documentation system.

## Costs and cost effectiveness analysis

Itemized cost menu was followed to estimate the cost of each activity by identifying cost components [3]. Cost components include cost items such as personnel, training, transport and supplies. The cost items in each cost component were identified and the cost of each item was calculated from the unit cost and the number of units utilized. Financial cost was calculated using the actual expenditure for each activity. Economic cost was calculated by adding opportunity cost for the time diverted by the study investigators for field supervision and training the project staff. Capital cost such as building, equipment such as microscope and transport were not considered. Finally, the per capita cost per sample was used to compare the costs between the strategies.

## Data analysis

Statistical software SPSS- 22.0 was used for analysis of data on prevalence and intensity. The critical cut off of 1% Mf prevalence, 2% CFA, 3 CFA positive children and 0.25% vector infection in each site was used for classifying the area as endemic or non-endemic. If any of the parameter is above the critical cut off in any of the site (site specific), the district is classified as endemic. Mosquito 'pool infection rate' was calculated as the percentage of pools positive for *W. bancrofti* DNA relative to the number of pools screened by individual cluster. Confidence intervals (CIs) were calculated using the Wilson method. Costs for different strategies were analyzed for cost components and different activities for one district (Balangir). The maps were generated using QGIS software version 3.4.13, which is an open-source software.

## Ethical statement

The study was approved by both the Institute Human Ethical Committee of the ICMR-Regional Medical Research Centre, Bhubaneswar (ICMR-RMRC/IHEC-2020/029) and Research & Ethics Committee of Department of Health and Family Welfare, Govt. Of Odisha (letter No.7033/MS-2-IV-04/2020(PT)dated 22/3/21) Survey teams explained the purpose of the study and study procedures to the selected household members and obtained written informed consent from adults and parents/legal guardians in case of children for conducting the tests in community surveys. District education officers of the study districts provided approval for

conducting CFA survey in school following which the written informed consent was obtained from the Headmaster/Headmistress of the school. Oral consent was obtained from the head of the households where Gravid traps were placed. All individuals who tested positive for filarial infection (Mf and CFA) were referred to the nearest hospital for treatment as per the guideline of NCVBDC.

## Results

The morbidity surveillance data available with NCVBDC-Odisha showed 1405 clinical cases (hydrocele: 1256, lymphedema: 149) in 439 villages/ wards in Kalahandi district. Clinical cases were ranging from 1 and 38 in different villages and wards. In Balangir district there were 3358 clinical cases (hydrocele: 2757 lymphedema: 601) in 759 villages/wards. Clinical cases were reported ranging from 1 and 56 in different villages and wards. As expected, hydrocele cases were predominant in both the districts.

### Purposive sampling

**Kalahandi district. Purposive sampling:** Two of the five sites selected for purposive sampling had less than 1000 population and hence a hamlet proximal to the village was included to reach the sample size. The reported cases were distributed as per the source NCVBDC, Accredited Social Health Activists (ASHAs) and household enumeration during the current study. The five sites were distributed in 5 different blocks (Fig 3). A total number of 11

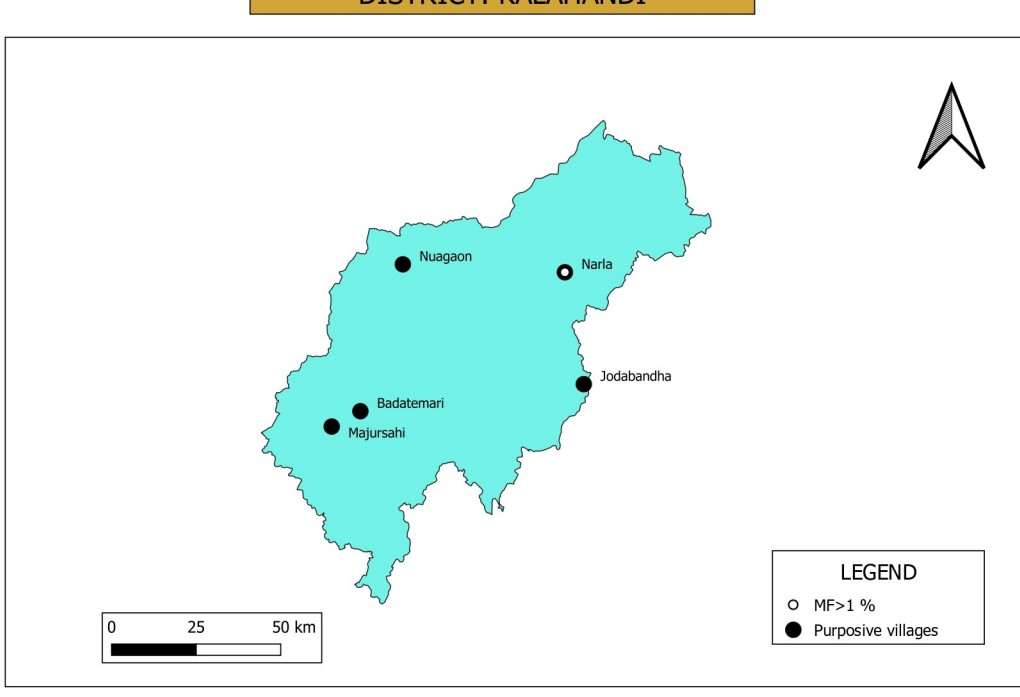

**Fig 3. Map showing location of purposive sampling villages in Kalahandi district.** Villages that have Mf prevalence above threshold value (>1%) during the night blood survey are marked in the figure. Map was downloaded from Balk, D., M. R. Montgomery, H. Engin, N. Lin, E. Major and B. Jones. 2020. Spatial Data from the 2011 India Census. Palisades, NY: NASA Socioeconomic Data and Applications Center (SEDAC). https://doi.org/10.7927/gya1-wp91. Accessed on Sept 11th 2023. No copyrighted material was used. The data was further modified using QGIS software version 3.4.13.

**Table 1. Source of data on clinical cases in the five purposive sampling villages of Kalahandi district.**

| CHC (Health Blocks) | Sub Centre | Village | Population | Hydrocele | | | Lymphodema | | | Total (Resesrch Data) |
|---|---|---|---|---|---|---|---|---|---|---|
| | | | | NCVBDC Data | ASHA's Data | Research Data | NCVBDC Data | ASHA's Data | Research Data | |
| Narla | Narla | Narla 1 | 1870 | 22 | 10 | 4 | 4 | 2 | 15 | 19 |
| Chapuria | Nuagaon | Nuagaon | 1050 | 10 | 0 | 2 | 0 | | 6 | 8 |
| Bnpur | Champadeipur | Jodabandha | 642 | 14 | 0 | 2 | 0 | 0 | 0 | 2 |
| | | Chandanpur | 799 | 0 | 0 | 0 | 0 | 0 | 0 | 0 |
| Koksara | Chikli | Majur Sahi | 824 | 14 | 2 | 3 | 0 | 0 | 0 | 3 |
| | | Gadaramala | 535 | 0 | 0 | 0 | 0 | 0 | 0 | 0 |
| Jaipatna | Badtemri | Badtemri | 1569 | 19 | 0 | 0 | 1 | 0 | 0 | 0 |
| | | Total | 7289 | 79 | 12 | 11 | 5 | 2 | 21 | 32 |

hydrocele and 21 lymphedema cases were observed in the 5 study sites during the household enumeration for the study. No filarial case was recorded in two hamlets and one village. Line listing cases by the program and available record with ASHAs showed higher number of cases, particularly hydrocele cases (Table 1). Number of individuals screened for Mf ranged from 458 to 641 in different villages. Mf positive cases were detected only in one village (3.3%) and the overall Mf prevalence was 0.61% (17/2782) and all are of only *W. bancrofti* parasite. Out of 17 positive cases, 13 were males (76%) (Table 2).

**Clustered random sampling**: Thirty clusters were selected in Kalahandi district as shown in Fig 4. While community household survey and MX were carried out in these clusters, Mini-TAS was conducted in the schools located in the selected clusters.

**Mini-TAS:** A total of 480 children (243 boys and 237 girls) were screened in 30 schools located in the study sites (Table 3). All children tested for CFA were using FTS living in the current location since from their birth. A total number of 8 children (Age: 11.21± 0.36 years) were detected antigen positive in five sites which is above the critical cutoff of 3. Amongst them 7 had FTS score one and one had FTS score 3.

**Community based household Mf survey:** The community-based household Mf survey was conducted in in 30 selected clusters of Kalahandi district. A total of 1989 individuals in the age group of 10 years and above (adults) were screened for microfilariae during night blood survey. Three individuals (0.15%) were Mf positive and all are of *W. bancrofti* (Table 4). All Mf positive cases were from one village and the overall Mf prevalence is less than 1%.

**Molecular Xenomonitoring:** A total of 156 households were selected from 30 sites of Kalahandi district using systematic sampling from the total number of households in the sites. A

**Table 2. Mf prevalence detected in five purposive sampling sites in Kalahandi district.**

| CHC (Health Blocks) | Sub Centre | Village | Total Slide Collected per village | MF +ve | Mf prevalence (%) |
|---|---|---|---|---|---|
| Narla | Narla | Narla 1 | 515 | 17 | 3.30 |
| Chapuria | Nuagaon | Nuagaon | 558 | 0 | 0 |
| Bnpur | Champadeipur | Jodabandha | 458 | 0 | 0 |
| | | Chandanpur | | | |
| Koksara | Chikli | Majur Sahi | 641 | 0 | 0 |
| | | Gadaramala | | | |
| Jaipatna | Badtemri | Badtemri | 610 | 0 | 0 |
| | | Total | 2782 | 17 | 0.61 |

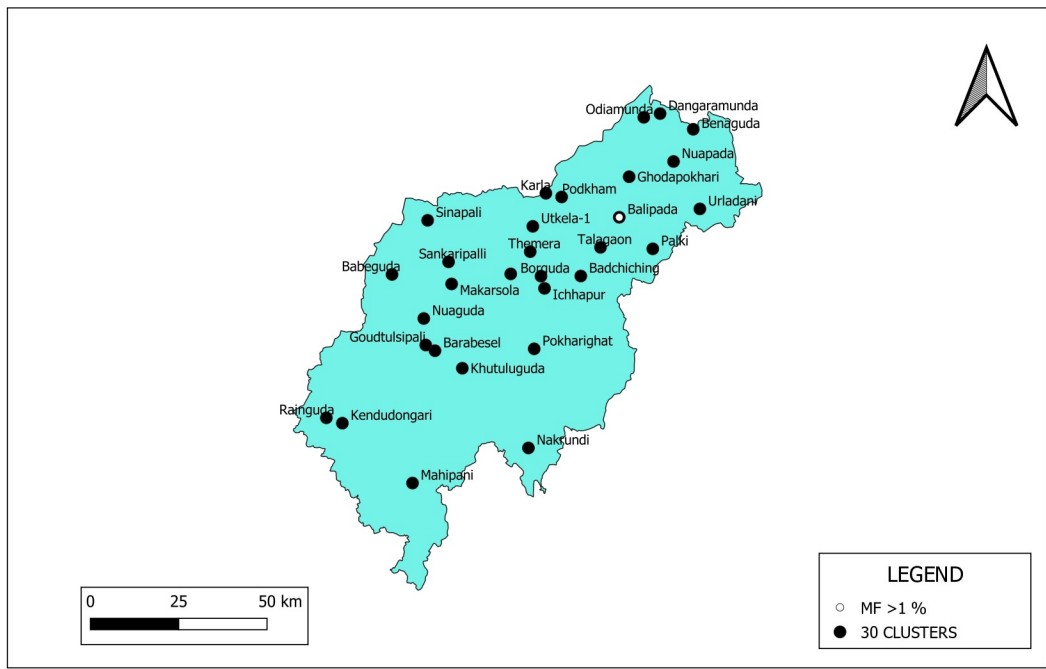

**Fig 4. Map showing location of 30 clusters selected for random sampling in Kalahandi district.** Only one out of 30 villages/wards showed Mf positive and above >1% during the household-based community survey. Map was downloaded from Balk, D., M. R. Montgomery, H. Engin, N. Lin, E. Major and B. Jones. 2020. Spatial Data from the 2011 India Census. Palisades, NY: NASA Socioeconomic Data and Applications Center (SEDAC). https://doi.org/10.7927/gya1-wp91. Accessed on Sept 11th 2023. No copyrighted material was used. The data was further modified using QGIS software version 3.4.13.

total of 7759 female mosquitoes were collected from the 156 households spread over 30 clusters. *Cx quinquefasciatus*, was the predominant mosquito species constituting about 97.98% of the total mosquitoes collected. *Anopheles* (1.22%) and *Aedes* (0.78%) species were rare (Fig 5A). As many as 320 pools were made from 7603 *Cx quinquefasciatus* gravid females. About 72% of the mosquitoes were gravid while 14% were half-gravid and 13,6% were freshly-fed (Fig 5B). However, none of the pools were found positive for *W. bancrofti* parasite DNA by PCR.

**Balangir district. Purposive sampling:** For the purposive sampling five sites including 2 hamlets were selected in Balangir district in the order of highest disease prevalence. As the population was lesser than 1000 in two sites, a hamlet at the proximity of the village was included (Fig 6). Enumeration of clinical cases showed 112 hydrocele and 18 lymphedema cases in these villages. All the villages including hamlets were recorded with clinical cases. The cases reported by ASHAs and program were lower except hydrocele cases reported by the program (Table 5). The disease prevalence in the five villages vary from 0.61% to 5.06%. Screening of 3453 individuals in these villages showed Mf positive cases with *W. bancrofti* parasite in all the villages including hamlets. Mf prevalence ranged from 4.6% to 10.3% in different villages and overall Mf prevalence was 8.2%. (Table 6). Age and gender specific analysis showed that prevalence was higher in males compared to females and prevalent only among adult population.

**Clustered random sampling:** Thirty clusters were selected in Balangir district as shown in Fig 7. Mini-TAS was conducted in the schools, while community-based household survey and MX were carried out in the selected clusters.

**Table 3. Results of Mini-TAS in Kalahandi district.**

| Sl. No. | CHC (Health Blocks) | SC | Village | Number screened for CFA | | Number positive | CFA score | | |
|---|---|---|---|---|---|---|---|---|---|
| | | | | **Boys** | **Girls** | | **1** | **2** | **3** |
| 1 | Parla | Tarapur | Ichhapur | 9 | 7 | 0 | 0 | 0 | 0 |
| 2 | Junagarh | Dundelmal | Borguda | 10 | 6 | 1 | 1 | 0 | 0 |
| 3 | Narla | Khairmal | Balipada | 8 | 8 | 0 | 0 | 0 | 0 |
| 4 | Borda | Karlaguda | Sankhairamal | 8 | 8 | 1 | 1 | 0 | 0 |
| 5 | Pastikudi | Utkela | Utkela-1 | 10 | 6 | 2 | 1 | 0 | 1 |
| 6 | Chapuria | Rengsapali | Sankripali | 9 | 7 | 0 | 0 | 0 | 0 |
| 7 | Narla | Santhpur | Badchiching | 10 | 6 | 0 | 0 | 0 | 0 |
| 8 | Chapuria | Chapuria | Babeguda | 8 | 8 | 0 | 0 | 0 | 0 |
| 9 | Junagarh | Banijora | Barabesel | 9 | 7 | 0 | 0 | 0 | 0 |
| 10 | Junagarh | Nandol | Nandol | 12 | 4 | 0 | 0 | 0 | 0 |
| 11 | Borda | Borbhatta | Nuapada | 6 | 10 | 0 | 0 | 0 | 0 |
| 12 | Pastikudi | Parlasingha | Podkhamb | 5 | 11 | 2 | 2 | 0 | 0 |
| 13 | Biswanathpur | Bengaon | Ghodaphokhri | 9 | 7 | 0 | 0 | 0 | 0 |
| 14 | Chapuria | Kegaon | Sinapali | 8 | 8 | 0 | 0 | 0 | 0 |
| 15 | Borda | Pokharighat | Pokharighat | 9 | 7 | 0 | 0 | 0 | 0 |
| 16 | Jaipatna | Uchhula | Nuaguda | 0 | 16 | 0 | 0 | 0 | 0 |
| 17 | Parla | Jayantpur | Rainguda | 8 | 8 | 0 | 0 | 0 | 0 |
| 18 | Kalampur | Bandhakana | Temera | 7 | 9 | 0 | 0 | 0 | 0 |
| 19 | Koksara | Kendudangari | Kendudangari | 10 | 6 | 0 | 0 | 0 | 0 |
| 20 | Karlamunda | Karlamunda | Odiamunda | 6 | 10 | 2 | 2 | 0 | 0 |
| 21 | Parla | Ainlajore | Khutuluguda | 8 | 8 | 0 | 0 | 0 | 0 |
| 22 | Junagarh | Maliguda | Goudtulsipali | 7 | 9 | 0 | 0 | 0 | 0 |
| 23 | Koksara | Badapodaguda | Talagaon | 9 | 7 | 0 | 0 | 0 | 0 |
| 24 | Jaipatna | Talguda | Mahipani | 10 | 6 | 0 | 0 | 0 | 0 |
| 25 | Pastikudi | Belkhandi | Karla | 8 | 8 | 0 | 0 | 0 | 0 |
| 26 | M.Rampur | Urladani | Dhobamunda | 10 | 6 | 0 | 0 | 0 | 0 |
| 27 | Karlamunda | Gajabahal | Dangaramunda | 7 | 9 | 0 | 0 | 0 | 0 |
| 28 | Th. Rampur | Nakrundi | Jhaudingjore | 12 | 4 | 0 | 0 | 0 | 0 |
| 29 | M.Rampur | Gochhadengen | Benaguda | 8 | 8 | 0 | 0 | 0 | 0 |
| 30 | Biswanathpur | Bhurti | Palki | 3 | 13 | 0 | 0 | 0 | 0 |
| | | | Total | 243 | 237 | 8 | 7 | 0 | 1 |

**Mini-TAS**: A total of 480 children (245 boys and 235 girls) with no history of migration in the age class 9–14 years were screened for antigenaemia by FTS in the schools in 30 clusters. A total number of 27 children (5.6%) were detected antigen positive in twelve random sites. Amongst them 12 had FTS score of 1 (44.4%), 8 had a FTS score of 2 (29.67%) and 7 had a FTS score of 3 (25.9%) (Table 7).

**Community Mf survey**: In Balangir district a total 2677 individuals were screened in 30 villages during community-based night blood survey. Number of clinical cases ranged from 3 to 25 different villages Microfilariae were detected in 14 villages and all had shown Mf prevalence of more than 1%. In all the 14 clusters detected with Mf positive cases were recorded with clinical cases. A total of 76 Mf positive individuals were detected in all 30 study sites and the overall prevalence is 2.9% (Table 8).

**Molecular Xenomonitoring**: A total 8285 mosquitoes were collected from selected 152 households spread over 30 clusters of Balangir district. *Cx quinquefasciatus* constituted about

**Table 4. Results of community Mf survey in Kalahandi district.**

| Sl. No. | CHC (Health Blocks) | SC | Village | Population | Number Examined | Number Mf +ve | Mf prevalence (%) |
|---|---|---|---|---|---|---|---|
| 1 | Parla | Tarapur | Ichhapur | 840 | 66 | 0 | 0 |
| 2 | Junagarh | Dundelmal | Borguda | 1784 | 84 | 0 | 0 |
| 3 | Narla | Khairmal | Balipada | 1400 | 88 | 3 | 3.41 |
| 4 | Borda | Karlaguda | Sankhairamal | 960 | 62 | 0 | 0 |
| 5 | Pastikudi | Uthkela | Utkela-1 | 1520 | 82 | 0 | 0 |
| 6 | Chapuria | Rengsapali | Sankaripalli | 623 | 42 | 0 | 0 |
| 7 | Narla | Shantapur | Badchiching | 505 | 43 | 0 | 0 |
| 8 | Chapuria | Chapuria | Babeguda | 803 | 41 | 0 | 0 |
| 9 | Junagarh | Banijara | Barabesel | 950 | 56 | 0 | 0 |
| 10 | Junagarh | Nandol | Nandol | 1119 | 93 | 0 | 0 |
| 11 | Borda | Borbhata | Nuapada | 826 | 57 | 0 | 0 |
| 12 | Pastikudi | Paralsinga | Podkham | 1145 | 66 | 0 | 0 |
| 13 | Biswanathpur | Bengaon | Ghodapokhari | 194 | 14 | 0 | 0 |
|  |  |  | Pajibahali | 367 | 23 | 0 |  |
| 14 | Chapuria | Kegaon | Sinapali | 1596 | 81 | 0 | 0 |
| 15 | Borda | Pokharighat | Pokharighat | 483 | 44 | 0 | 0 |
| 16 | Jaipatna | Uchhala | Nuaguda | 840 | 96 | 0 | 0 |
| 17 | Parla | Jayantpur | Rainguda | 887 | 71 | 0 | 0 |
| 18 | Kalampur | Bandhkana | Temera | 2405 | 121 | 0 | 0 |
| 19 | Koksara | Kendudongri | Kendudungri | 1122 | 68 | 0 | 0 |
| 20 | Karlamunda | Karlamunda | Odiamunda | 1065 | 101 | 0 | 0 |
| 21 | Parla | Ainlajore | Khutuluguda | 1000 | 68 | 0 | 0 |
| 22 | Junagarh | Maliguda | Goudtulsipali | 832 | 73 | 0 | 0 |
| 23 | Koksara | Badapodaguda | Talagaon | 865 | 57 | 0 | 0 |
| 24 | Jaipatna | Talaguda | Mahipani | 711 | 62 | 0 | 0 |
| 25 | Pastikudi | Belkhandi | Karla | 460 | 39 | 0 | 0 |
|  |  |  | Singaribahal | 430 | 35 | 0 |  |
| 26 | M. Rampur | Urladani | Dhobamunda | 892 | 56 | 0 | 0 |
| 27 | Karlamunda | Gajabahal | Dangaramunda | 276 | 48 | 0 | 0 |
|  |  |  | Gajabahal | 579 |  | 0 |  |
| 28 | Th. Rampur | Nakrudni | Jhaudingjore | 420 | 68 | 0 | 0 |
| 29 | M. Rampur | Gochhadengen | Benaguda | 548 | 42 | 0 | 0 |
| 30 | Biswanathpur | Bhurti | Palki | 246 | 42 | 0 | 0 |
|  |  |  | Bhurtigarh | 280 |  | 0 |  |
|  | Total |  |  | 28973 | 1989 | 3 | 0.15 |

96% of the total mosquitoes collected and Anopheles, *Armingeris* and *Aedes* constituted only 1.6%, 1.4% and 0.78% respectively (Fig 5A). A total of 7990 *Cx. quinquefasciatus* gravid females were collected from 152 households and 300 pools were made. Among them 81.75%, 12.81%, and 5.43% of were gravid, half-gravid, and freshly-fed respectively (Fig 5B). *Cx. quinquefasciatus* gravid mosquitoes collected from 152 HHs were processed for *W. bancrofti* infection. Out of 300 pools, 10 pools (3.0%) were found positive for *W. bancrofti* infection by PCR. Percentage of mosquitoes positive for parasite DNA was 3.12% and vector infection was found in 3 clusters. In Balangir both the sampling strategies and all the indicators were above the threshold and hence the district is endemic.

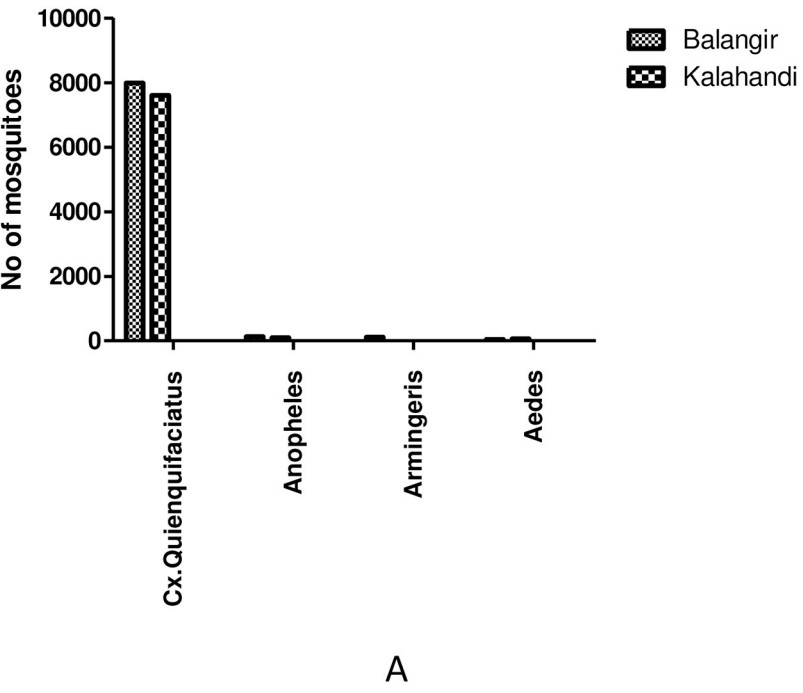

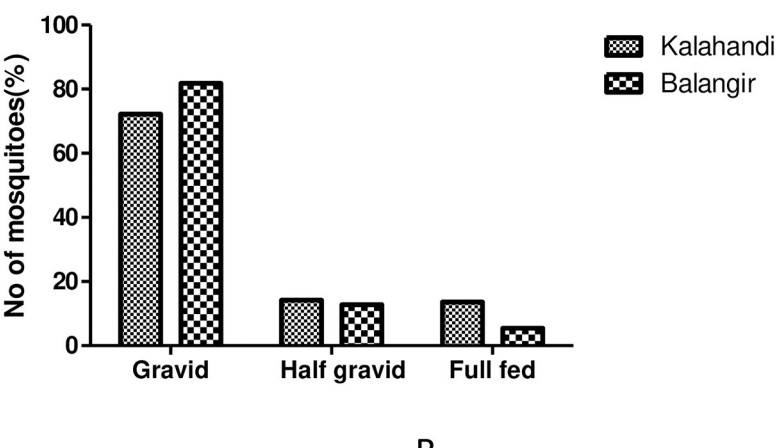

**Fig 5.** A. Prevalence of mosquitoes in study districts. B. Abdominal condition of mosquitoes in study districts.

## Correlation matrix analysis between community Mf survey, Mini-TAS and MX

Correlation matrix analysis was done between Mini-TAS, Mf and mosquito pool infection rate in the random sampling method. In Kalahandi district, we did not find any significant relationship between Mf positivity, FTS positive and mosquito pool infection rate. However

**Fig 6. Map showing purposive sampling villages in Balangir district.** Map was downloaded from Balk, D., M. R. Montgomery, H. Engin, N. Lin, E. Major and B. Jones. 2020. Spatial Data from the 2011 India Census. Palisades, NY: NASA Socioeconomic Data and Applications Center (SEDAC). https://doi.org/10.7927/gya1-wp91. Accessed on Sept 11th 2023. No copyrighted material was used. The data was further modified using QGIS software version 3.4.1.

a significant strong relationship between Mf positivity and FTS (CFA) positive rate ($R^2 = 0.636$, $P<0.01$) was observed in Balangir district (Fig 8). Comparison between random and purposive sampling shows that the critical cut-off value of Mf prevalence, CFA positive children and vector infection was higher in the Balangir district in every sampling strategy but Mini-TAS is only above the threshold value in Kalahandi district as depicted in Table 9.

**Table 5. Different sources of clinical cases in the five purposive sampling villages of Balangir district.**

| CHC (Health Block) | Sub Centre | Village | Population | Hydrocele | | | Lymphodema | | | Total (Research Data) |
|---|---|---|---|---|---|---|---|---|---|---|
| | | | | NCVBDC Data | ASHA's Data | Research Data | NCVBDCP Data | ASHA's Data | Research Data | |
| Jamgaon | Mehermunda | Brahamanipali | 830 | 49 | 11 | 23 | 0 | 2 | 19 | 42 |
| Jamgaon | Chhatamakha | Belbahali | 430 | 13 | 0 | 1 | 5 | 0 | 8 | 9 |
| | | Laderbahal | 1088 | 4 | 0 | 9 | 0 | 2 | 2 | 11 |
| Loisingha | Hirapur | Tevadungri | 528 | 18 | 0 | 2 | 0 | 2 | 0 | 2 |
| | | Gopalpur | 779 | 13 | 5 | 4 | 0 | 0 | 3 | 6 |
| Loisingha | Budhipadar | Budhipadar | 1150 | 17 | 6 | 22 | 3 | 2 | 0 | 22 |
| Loisingha | Kusmel | Kusmel | 2906 | 24 | 9 | 51 | 0 | 1 | 0 | 51 |
| | | Total | 7711 | 138 | 31 | 112 | 8 | 9 | 32 | 144 |

**Table 6. Mf prevalence in five purposive sampling sites in Balangir district.**

| CHC (Health Blocks) | SC | Villages | Total House-holds | No. of Slide Collected | Total number of slide collected per village | Mf Positive | Mf prevalence (%) |
|---|---|---|---|---|---|---|---|
| Jamgaon | Mehermunda | Brahamanipali | 236 | 632 | 632 | 64 | 10.1 |
| Jamgaon | Chhatamakha | Belbahali | 79 | 202 | 910 | 19 | 10.3 |
| | | Laderbahal | 275 | 708 | | 75 | |
| Loisingha | Hirapur | Tevadungri | 118 | 366 | 886 | 17 | 4.6 |
| | | Gopalpur | 185 | 520 | | 24 | |
| Loisingha | Budhipadar | Budhipadar | 307 | 559 | 559 | 33 | 7.1 |
| Loisingha | Kusmel | Kusmel | 700 | 466 | 466 | 51 | 9.1 |
| | | Total | | | 3453 | 283 | 8.2 |

## Cost analysis

Costs of different sampling strategies by cost component and activities are shown in Tables 10 and 11. Purposive sampling and different survey methods followed in random sampling are shown in Table 10. Comparison of per capita cost showed, purposive sampling is least expensive costing about INR 0.09. All the other strategies costs almost double the cost of purposive sampling. The cost per sample was least with purposive sampling and Mini-TAS was very expensive. The cost of a pool mosquitoes and molecular assay was as high as INR 812. Analysis of data by

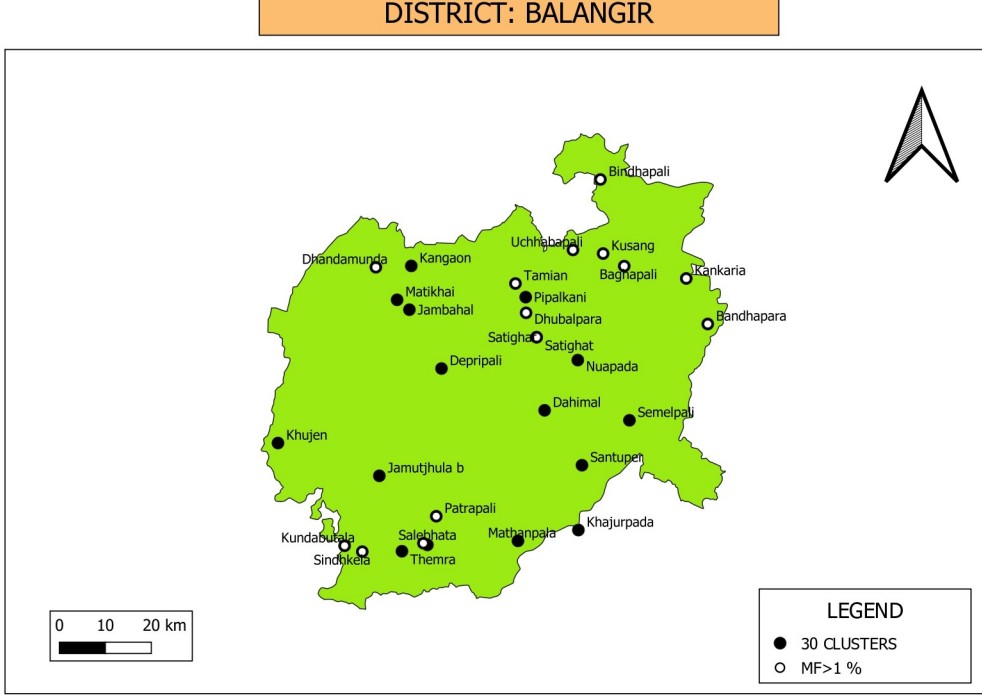

**Fig 7. Map showing location of 30 clusters selected for random sampling in Balangir district.** Fourteen villages/ wards showed Mf prevalence more than the threshold value (>1%) during the household-based community survey. Map was downloaded from Balk, D., M. R. Montgomery, H. Engin, N. Lin, E. Major and B. Jones. 2020. Spatial Data from the 2011 India Census. Palisades, NY: NASA Socioeconomic Data and Applications Center (SEDAC). https://doi.org/10.7927/gya1-wp91. Accessed on Sept 11th 2023. No copyrighted material was used. The data was further modified using QGIS software version 3.4.13.

**Table 7. Results of Mini-TAS in Balangir district.**

| Sl. No. | CHC (Health Blocks) | SC | Village | Number screened for CFA | | Number positive | CFA score | | |
|---|---|---|---|---|---|---|---|---|---|
| | | | | Boys | Girls | | 1 | 2 | 3 |
| 1 | Chudapali | Manhira | Baghapali | 6 | 10 | 6 | 2 | 2 | 2 |
| 2 | Loisingha | Kusang | Kusang | 11 | 5 | 5 | 3 | 1 | 1 |
| 3 | Loisingha | Uparbahal | Uchhabapali | 12 | 4 | 0 | 0 | 0 | 0 |
| 4 | Deogaon | Ramchandrapur | Satighat | 6 | 10 | 1 | 0 | 0 | 1 |
| 5 | Saintala | Budabahal | Khajurpada | 7 | 9 | 1 | 1 | 0 | 0 |
| 6 | Agalpur | Salebhata | Salebhata | 5 | 11 | 2 | 1 | 0 | 1 |
| 7 | Chudapali | Bhundimuhan | Pipalkani | 6 | 10 | 1 | 1 | 0 | 0 |
| 8 | Agalpur | Bharsuja | Bindhapali | 11 | 5 | 0 | 0 | 0 | 0 |
| 9 | Ghasian | Tamian | Tamian | 8 | 8 | 3 | 1 | 1 | 1 |
| 10 | Belapada | Mandal | Depripali | 5 | 11 | 0 | 0 | 0 | 0 |
| 11 | Ghasian | Bhainsa | Dhubalpara | 14 | 2 | 1 | 0 | 1 | 0 |
| 12 | Ghasian | Jogimunda | Jambahal | 10 | 6 | 0 | 0 | 0 | 0 |
| 13 | Jamgaon | Bhaler | Kankaria | 9 | 7 | 3 | 1 | 1 | 1 |
| 14 | Gudvella | Nuapada | Nuapada | 6 | 10 | 0 | 0 | 0 | 0 |
| 15 | Khaprakhol | Dhandamunda | Dhandamunda | 5 | 11 | 2 | 1 | 1 | 0 |
| 16 | Jamgaon | Mahimunda | Dahimal | 13 | 3 | 1 | 0 | 1 | 0 |
| 17 | Khaprakhol | Padiabahal | Kangaon | 1 | 15 | 0 | 0 | 0 | 0 |
| 18 | Kholan | Banjipadar | Mathanpala | 9 | 7 | 0 | 0 | 0 | 0 |
| 19 | Turekela | Ghunesh | Khujen | 6 | 10 | 0 | 0 | 0 | 0 |
| 20 | Kholan | Kursud | Sukunabhata | 10 | 6 | 0 | 0 | 0 | 0 |
| 21 | Deogaon | Bandhapara | Bandhapara | 7 | 9 | 0 | 0 | 0 | 0 |
| 22 | Muribahal | Haldi | Semelpali | 8 | 8 | 0 | 0 | 0 | 0 |
| 23 | Saintala | Kuargaon | Santuper | 10 | 6 | 0 | 0 | 0 | 0 |
| 24 | Sindhekela | Alanda | Themra | 5 | 11 | 0 | 0 | 0 | 0 |
| 25 | Sindhekela | Sindhekela | Sindhkela | 10 | 6 | 0 | 0 | 0 | 0 |
| 26 | Belpada | Gambhari | Matikhai | 9 | 7 | 0 | 0 | 0 | 0 |
| 27 | Turekela | Dhamandanga | Jamutjhulab | 8 | 8 | 0 | 0 | 0 | 0 |
| 28 | Sindhekela | Chulifunka | Kundabutala | 11 | 5 | 0 | 0 | 0 | 0 |
| 29 | Kholan | Sanapatrapali | Satighat | 7 | 9 | 0 | 0 | 0 | 0 |
| 30 | Muribahal | Patrapali | Patrapali (Kha) | 10 | 6 | 1 | 1 | 0 | 0 |
| | | Total | | 243 | 245 | 27 | 12 | 8 | 7 |

cost component showed that personal cost was the major cost component (60.2%) for purposing sampling while it was transport (63.2%) for community Mf survey and supplies (57.6%) for Mini-TAS. The costs of supplies and transport constituted 46.2% and 43.1% of the total cost for MX.

Cost comparison by activity showed that cost for slide processing and examination was the major cost intensive activity constituting about 45.4% of the total cost for purposive sampling. The major resource intensive activity was blood collection and testing for Mini-TAS (77.2%) while it was (46.2%) for MX. The costs for all related activities for community Mf survey were comparable.

## Discussion

The World Health Organization has recently proposed 2030 as the new target year for elimination of LF as a public health problem [12], whereas India is committed to eliminate LF by 2027

**Table 8. Results of community Mf survey in Balangir district.**

| Sl. No. | CHC (Health Blocks) | SC | Village | Population | No. examined | Mf +ve (n) | Mf prevalence (%) |
|---|---|---|---|---|---|---|---|
| 1 | Chudapali | Manhira | Baghapali | 1413 | 63 | 4 | 6.3 |
| 2 | Loisingha | Kusang | Kusang | 3313 | 133 | 11 | 8.4 |
| 3 | Loisingha | Uparbahal | Uchhabapali | 557 | 36 | 1 | 2.8 |
| 4 | Deogaon | Ramchandrapur | Satighat | 947 | 75 | 0 | 0 |
| 5 | Saintala | Budabahal | Khajurpada | 774 | 77 | 0 | 0 |
| 6 | Agalpur | Salebhata | Salebhata | 4082 | 165 | 21 | 12.7 |
| 7 | Chudapali | Bhundimuhan | Pipalkani | 527 | 29 | 0 | 0 |
| 8 | Agalpur | Bharsuja | Bindhapali | 755 | 59 | 1 | 1.7 |
| 9 | Ghasian | Tamian | Tamian | 3070 | 184 | 15 | 8.2 |
| 10 | Belapada | Mandal | Depripali | 1472 | 86 | 0 | 0 |
| 11 | Ghasian | Bhainsa | Dhubalpara | 1360 | 83 | 3 | 3.6 |
| 12 | Ghasian | Jogimunda | Jambahal | 1692 | 84 | 0 | 0 |
| 13 | Jamgaon | Bhaler | Kankaria | 603 | 51 | 3 | 5.9 |
| 14 | Gudvella | Nuapada | Nuapada | 2283 | 119 | 0 | 0 |
| 15 | Khaprakhol | Dhandamunda | Dhandamunda | 1885 | 90 | 1 | 1.1 |
| 16 | Jamgaon | Mahimunda | Dahimal | 485 | 43 | 0 | 0 |
| 17 | Khaprakhol | Padiabahal | Kangaon | 751 | 34 | 0 | 0 |
| 18 | Kholan | Banjipadar | Mathanpala | 890 | 53 | 0 | 0 |
| 19 | Turekela | Ghunesh | Khujen | 1197 | 66 | 0 | 0 |
| 20 | Kholan | Kursud | Sukunabhata | 590 | 37 | 0 | 0 |
| 21 | Deogaon | Bandhapara | Bandhapara | 906 | 48 | 1 | 2.9 |
| 22 | Muribahal | Haldi | Semelpali | 415 | 44 | 0 | 0 |
| 23 | Saintala | Kuargaon | Santuper | 833 | 75 | 0 | 0 |
| 24 | Sindhekela | Alanda | Themra | 1805 | 81 | 0 | 0 |
| 25 | Sindhekela | Sindhekela | Sindhkela | 6514 | 418 | 12 | 2.9 |
| 26 | Belpada | Gambhari | Matikhai | 1250 | 71 | 0 | 0 |
| 27 | Turekela | Dhamandanga | Jamutjhula b | 2028 | 106 | 0 | 0 |
| 28 | Sindhekela | Chulifunka | Kundabutala | 1736 | 99 | 1 | 1.0 |
| 29 | Kholan | Sanapatrapali | Satighat | 956 | 76 | 1 | 1.3 |
| 30 | Muribahal | Patrapali | Patrapali | 1670 | 92 | 1 | 1.1 |
| | Total | | | | 2677 | 76 | 2.8 |

three years ahead of the global target [13]. However, for successful elimination of LF at national level and to get the validation, it is necessary to ascertain that non-MDA districts are not endemic for LF with no active transmission by conducting mapping (remapping or confirmatory mapping). Till 2022, only 17 the total countries out of 72 endemic countries have met these criteria and declared as LF free after meeting these criteria [14]. India, therefore, needs to assess the current status of LF in the districts with unsurveyed/ uncertain LF endemicity at an earliest so as to provide the information in the dossier. For this there is a pressing need to pin down an appropriate re-mapping tool to identify new transmission hotspots.

The WHO protocol uses two-stage sampling (cluster and convenience) for mapping LF in areas with uncertain endemicity by selecting two sites per district. In each site a convenience sample of 100 adults are tested for antigenemia or microfilaremia. One or more confirmed positive tests in either site is interpreted as an indicator of potential transmission, prompting MDA at the district-level [15]. While this mapping strategy has worked well in high-prevalence settings, imperfect diagnostics and the transmission potential of a single positive adult have

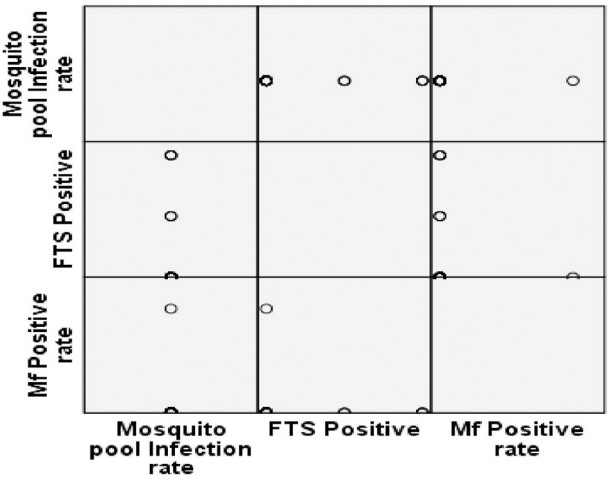

A

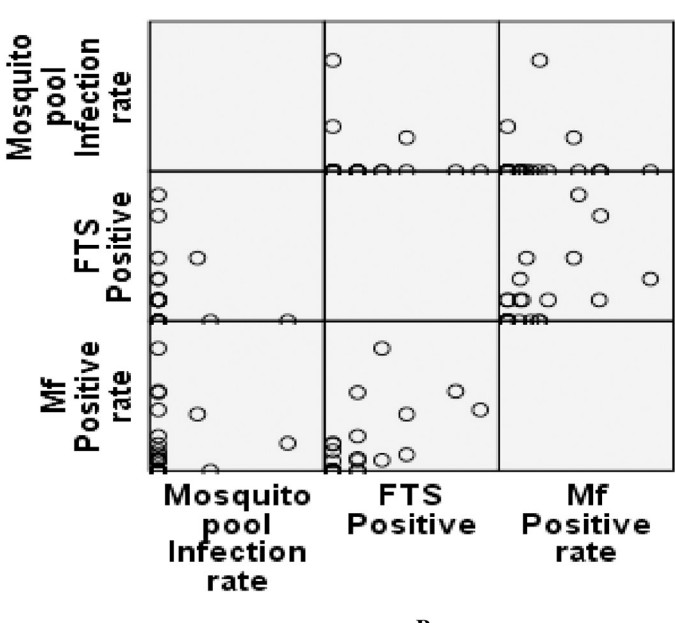

B

**Fig 8. Correlation matrix between Mf, FTS and mosquito pool infection in Kalahandi (A) and Balangir (B) district.**

**Table 9. Comparison between purposive sampling and random sampling critical cut-off value.**

| District | Purposive sampling | Random sampling | | |
|---|---|---|---|---|
| | Mf prevalence (critical cut off 1%) | Mini-TAS critical cut off 3 | Mf prevalence (critical cut off 1%) | vector infection mx survey (Critical cut off 0.25%) |
| Kalahandi | 0.61 | 8 | 0.150 | 0 |
| Balangir | 8.2 | 27 | 2.8 | 3.0 |

**Table 10. Financial cost (INR) of different sampling strategies in Balangir district by cost component.**

| Cost component (INR) | Purposive | Random | | |
|---|---|---|---|---|
| | | Mf | Mini-TAS | Mx |
| Personal | 86125 | 84500 | 29250 | 26000 |
| Transport | 43750 | 157500 | 105000 | 105000 |
| Supplies | 13250 | 6840 | 182000 | 112500 |
| Total | 143125 | 248840 | 316250 | 243500 |
| Per capita cost (INR) | 0.09 | 0.15 | 0.19 | 0.15 |
| Sample/mosquito pools | 3453 | 2677 | 480 | 300 |
| Cost per sample/pool of mosquitoes | 41.45 | 92.95 | 658.85 | 811.67 |

raised concerns about the strategy's use in low-prevalence settings. In response to these limitations, a statistically rigorous confirmatory mapping strategy was designed as a complement to the current strategy where LF endemicity is uncertain. In this strategy, schools are selected by either systematic or cluster sampling, depending on population size, and within each selected school, children 9–14 years are sampled systematically. All selected children are tested and the number of positive results is compared against a critical value to determine, with known probabilities of error, whether the average prevalence of LF infection is likely below a threshold of 2%. [6]. India is adopting the method of surveying human population for LF infection in areas suspected of having high risk transmission as an accelerated plan for identification of new endemic districts and recommendation of MDA [16]. This has helped in identifying a number of new districts for inclusion in MDA later on [17]. However, this protocol of identifying survey sites based primarily on the presence of lymphedema patients might be misleading in low transmission areas.

In the present study, in Balangir district, screening of 500 individuals in each of the purposive sampling site showed that all the five sites were above threshold value. Similarly, Mini-TAS, community Mf survey and MX in random sites showed above the thresholds confirming the endemic status of the district. The disease prevalence in the district as per the program report was 1.89% and the present study showed disease prevalence of 1.86%, ranging from 0.61% to 5.06% in different sites. Under such situation, purposive sampling is sensitive enough to confirm the endemicity. The cost of purposive sampling is also relatively lesser compared to random sampling. On the contrary, in Kalahandi district 4 out of 5 villages, selected purposively based on morbidity data, were found negative for Mf during night blood survey. The overall disease prevalence is 0.44% and varies from 0 to 1.02% in the five selected villages during purposive sampling. One site with a disease rate of 1.02% showed Mf prevalence 3.03%

**Table 11. Financial cost (INR) of different sampling strategies in Balangir district by activity.**

| Activity | Purposive | Random | | |
|---|---|---|---|---|
| | | Mf | Mini-TAS | Mx |
| Household enumeration | 20750 | 72000 | 0 | 0 |
| Preparatory visit | 10375 | 62250 | 62250 | 62250 |
| Mosquito collection | 0 | 0 | 0 | 9750 |
| Blood collection and testing for Mf/Ag | 47000 | 69090 | 244250 | 52500 |
| Slide processing and examination | 65000 | 45500 | 9750 | 6500 |
| Molecular assay of mosquito samples | 0 | 0 | 0 | 112500 |
| Total | 143125 | 248840 | 316250 | 243500 |

which is more than 1%. Similarly, in the random site even though one of the site showing Mf prevalence (as high as 3.4%) in a sample size of 88 with a disease rate of 0.07%, the overall Mf prevalence was less than 1%, indicative of non-endemic status of the district. However, Mini-TAS showed antigenemia prevalence more than the threshold (>3) confirming LF endemicity in this district.

The WHO recommended Mini-TAS assess the status of antigenemia among older children who have a longer period of potential exposure to LF infection to check the endemicity of LF in areas with uncertain endemicity, which has been already addressed during the field evaluation in Ethiopia and Tanzania [6,7]. Since the present study showed that the CFA positivity in children is above the critical cut-off of 3 in both the districts, hence confirms that both the districts are endemic and requiring MDA. Purposive sampling showed only one site above the threshold of 1% Mf prevalence in Kalahandi district. This method can be considered as less sensitive than Mini-TAS in areas with low disease prevalence. However, defining a threshold for disease prevalence may be challenging due to clustering of risk factors. Mini-TAS has only smaller sample but with quota sampling it may be possible to identify clusters with active transmission (hotspots) and this will be useful for follow-up in case necessary. The data from Mini-TAS can be used for identifying risk areas using geostatistical model [2]. This tool drastically not only decreased the estimated number of people at risk for LF transmission that required MDA but also had major impact on resource and logistic implications required for LF elimination program [7]. During the crosssectional household survey for Mf, 14 out of 30 sites showed Mf prevalence with an overall prevalence of 2.8% in Balangir district indicating high risk of transmission in these villages. Concordance between Mini-TAS and Mf in random sites in detecting positive sites in 10 out of 30 clusters suggested either antigenemia in children or Mf can be a good indicator of infection.

MX (the detection of filarial DNA in mosquitoes) is a potentially useful and has been recognized as a tool complementary to TAS for monitoring recrudescence of infection in post- and validation phases of LF elimination programs [18,19]. Present study showed that from both Mini-TAS and Mf-surveys confirmed the presence of Mf carriers. MX did not show any vector infection in Kalahandi district, where household survey showed Mf positive cases. The lack of concordance between MX and Mf-surveys at the cluster level could either be due to migration of individuals from endemic districts or the difference in the location of the HHs selected for the surveys. Lack of concordance has been reported elsewhere where both the surveys were carried out in the same HHs [20]. The non-invasive feature of MX has an edge over other methods and may be potentially provide a more sensitive measure when the infection is at a level lower than that detectable by anitigen or Mf-testing [20,21]. MX also did not show any agreement with Mini-TAS may be due to young children could be less exposed to mosquitoes than adults and children may be more often protected by bed nets when they sleep than adults [22,23] might cause disagreement between these two parameters. The limitation of MX is that it may not reflect the risk in term of infected individuals. Further, this laboratory intensive tool requires technical expertise to identify mosquitoes, assays and laboratory facilities for processing the samples in the health system [20]. However, a strong correlation found between Mf survey and Mini-TAS survey ($R^2$ = 0.636, P<0.01) in Balangir district, indicating the risk of infection and transmission.

Despite a number of studies on feasibility in terms of availability of adequate laboratory facilities and specially trained personnel for MX have been undertaken, yet the assessment of cost of the use of Mini-TAS and MX in different operational settings is required. MX is a non-invasive and more sensitive tool than school or community-based TAS in detecting residual hotspots, hence could be a promising tool for remapping and monitoring post MDA and validation phases [20]. In our study the cost of Mini-TAS was Rs. 3,16, 250.00 per EU, whereas

MX was Rs 2,43,500.00 and thirty clustered random Mf survey was Rs 2, 48,840.00. Though the cost of Mini-TAS during the study was higher than MX and clustered random Mf survey, but it is lower than that derived for Ehiopia (Rs 6,32,800.00) and Tanzania (Rs 7,67,840.00) [6]. Personnel and travel were found to be the major cost drivers in this study, as found in earlier by others [24]. One of the limitations of the costing analysis is that our study was conducted in a research mode, which may not reflect the costing under program, where there is scope for utilizing the existing manpower after appropriate training. Notwithstanding, the confirmatory mapping tool (Mini-TAS) is more expensive and resource-intensive than other approaches, however in a long run it is highly cost-effective since it has the ability to avoid unnecessary MDA [6]. For NTD programs, the ability to avoid unnecessary MDA not only saves precious financial resources and timely achievement of the goal of elimination, but also provides scope for utilizing more time, energy and human capital for activities where it is needed the most.

## Conclusion

India has established a strong base for achieving the goal of LF elimination. However, effective implementation of all activities including confirming the endemicity status of non-MDA districts is critical to achieve the target. The present study showed that the national guideline for remapping districts with uncertain endemicity is operationally feasible and cost effective, but sensitive enough in high disease prevalence situation. But particularly in low prevalence settings, Mini-TAS can benefit the national LF elimination programme in identifying the areas that require the intervention, thereby saving time, resources, man-power and avoid unnecessary treatments.

## Acknowledgments

The authors are thankful to Director Public Health and NVBDCP for their administrative support, and Chief Medical Officer, VBD consultant of Kalahandi and Balangir district for the cooperation and necessary support extended during the study. We would like to acknowledge the Head Masters and Teachers of concerned Schools for their assistance in conducting the Mini-TAS and parents who consented to their children's participation in the study. The authors are grateful to all stakeholders who participated in our study area for their cooperation during the night blood survey and also for placing the gravid traps on their household premises. We thank Debabrata Jena, and Dr. Jyoti Ghosal for their help with creating the Maps using QGIS software.

## Author Contributions

**Conceptualization:** Kaliannagounder Krishnamoorthy, Manoranjan Ranjit, Madhusmita Bal.

**Data curation:** Barsa Baisalini Panda, Arundhuti Das, Hitesh Kumar Jain, Sujata Dixit, Nilam Somalkar, Shubhashisha Mohanty, Manoranjan Ranjit, Madhusmita Bal.

**Formal analysis:** Barsa Baisalini Panda, Kaliannagounder Krishnamoorthy, Arundhuti Das, Manoranjan Ranjit, Madhusmita Bal.

**Funding acquisition:** Kaliannagounder Krishnamoorthy, Manju Rahi, Sanghamitra Pati, Manoranjan Ranjit, Madhusmita Bal.

**Investigation:** Barsa Baisalini Panda, Nilam Somalkar, Shubhashisha Mohanty, Manoranjan Ranjit, Madhusmita Bal.

**Methodology:** Kaliannagounder Krishnamoorthy, Manoranjan Ranjit, Madhusmita Bal.

**Project administration:** Manoranjan Ranjit, Madhusmita Bal.

**Resources:** Nilam Somalkar, Shubhashisha Mohanty, Manoranjan Ranjit, Madhusmita Bal.

**Supervision:** Kaliannagounder Krishnamoorthy, Manju Rahi, Sanghamitra Pati, Manoranjan Ranjit, Madhusmita Bal.

**Writing – original draft:** Barsa Baisalini Panda, Manoranjan Ranjit, Madhusmita Bal.

**Writing – review & editing:** Kaliannagounder Krishnamoorthy, Manju Rahi, Sanghamitra Pati, Manoranjan Ranjit, Madhusmita Bal.

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
