## [Decision Letter · Decision Letter 0]

22 Aug 2023

PONE-D-23-20937Mini-TAS as a confirmatory mapping tool for remapping areas with uncertain Filarial endemicity to exclude/ include for Mass Drug Administration: A report from field validation in IndiaPLOS ONE

Dear Dr. Bal,

Thank you for submitting your manuscript to PLOS ONE. After careful consideration, we feel that it has merit but does not fully meet PLOS ONE’s publication criteria as it currently stands. Therefore, we invite you to submit a revised version of the manuscript that addresses the points raised during the review process.

We look forward to receiving your revised manuscript.

Kind regards,

Dziedzom Komi de Souza, Ph.D.

Academic Editor

PLOS ONE

2. We note that Figures 1, 3, 4, 6, and 7 in your submission contain [map/satellite] images which may be copyrighted. All PLOS content is published under the Creative Commons Attribution License (CC BY 4.0), which means that the manuscript, images, and Supporting Information files will be freely available online, and any third party is permitted to access, download, copy, distribute, and use these materials in any way, even commercially, with proper attribution. For these reasons, we cannot publish previously copyrighted maps or satellite images created using proprietary data, such as Google software (Google Maps, Street View, and Earth). For more information, see our copyright guidelines: http://journals.plos.org/plosone/s/licenses-and-copyright.

1. You may seek permission from the original copyright holder of Figures 1, 3, 4, 6, and 7 to publish the content specifically under the CC BY 4.0 license.  

Reviewers' comments:

Reviewer's Responses to Questions

**Comments to the Author**

1. Is the manuscript technically sound, and do the data support the conclusions?

Reviewer #1: Yes

Reviewer #2: Yes

Reviewer #3: Yes

2. Has the statistical analysis been performed appropriately and rigorously? 

Reviewer #1: Yes

Reviewer #2: Yes

Reviewer #3: Yes

3. Have the authors made all data underlying the findings in their manuscript fully available?

Reviewer #1: Yes

Reviewer #2: Yes

Reviewer #3: Yes

4. Is the manuscript presented in an intelligible fashion and written in standard English?

Reviewer #1: Yes

Reviewer #2: Yes

Reviewer #3: Yes

5. Review Comments to the Author

Reviewer #1: Thank you for the opportunity to review the manuscript with topic; Mini-TAS as a confirmatory mapping tool for remapping areas with uncertain Filarial endemicity to exclude/ include for Mass Drug Administration: A report from field validation in India

This is a well-written manuscript. It is a critical topic in the history of NTDs and addresses one of the critical challenges of the WHO NTD Roadmap 2030. Done in a logical manner that makes it easy to follow and comprehend. It answers an important question critical to the endgame challenges of lymphatic filariasis elimination.

This is a decision-making framework. This study compares two mapping protocols to identify more sensitive and cost-effective mapping methods. The study also took the opportunity to assess the feasibility of alternative methods involving mosquito vector and the human adult populations.

The abstract is succinct and on point and bears a lot of relevance to the NTD community. Considering the 2030 milestones and targets of the WHO NTD Roadmap makes this study time sensitive and critical to the decision making process for India and other LF endemic countries. Even more critical India has set a timeline of 2027 for elimination of lymphatic filariasis from India. It should however be noted and placed in the right perspective that WHO NTD Roadmap has estimated that 81% and not 100% of LF endemic countries would have been validated as having eliminated lymphatic filariasis as a public health problem globally.

The manuscript is strengthened by taking each of the surveys conducted in isolation and comparing the results to arrive at its conclusions and recommendations. The sampling methods applied are cost-effective and feasible to undertake under programmatic conditions. In addition, the methods described in the manuscript are adaptable to the local country settings. This strengthens the paper as a source of guidance to that process. Therefore, the processes and methods suggested in this paper provide a framework for adaptation by the WHO and programs.

The statistical analysis is well-done and easy to replicate in tandem with WHO mini-TAS guidelines. The discussion is also to the point adds value to the manuscript. It leads to valid conclusions and therefore recommendations. However, considering the objective of this manuscript, clarity should be provided on the recommendations and way-forward for these findings.

The manuscript is very useful and relevant to the current strategies of the WHO NTD 2030 Roadmap and on elimination of lymphatic filariasis as a public health problem. The conclusions, lessons learned and recommendations will be of immense assistance to the normative functions and guidelines development of the WHO for the global programme for elimination of lymphatic filariasis.

The is a well-written manuscript. I recommend that it for publication after a few errors detected in the manuscript are corrected.

Reviewer #2: Overall observation

1. All sections of the manuscript is well written with the necessary information and data for each section.

2. Abstract is concise, clear and summarizes the research question, method of the study, high level results and conclusions supported by the results presented.

3. Introduction presents context, clear research questions and rationale for the study.

4. The method is clearly described and appropriate to address the research questions. Fig. 2 is an important summary of the method and great help to the reader. The authors have separated the various sampling arms of the study in the method, results and discussion sections making it easy to follow the study and findings.

5. Findings: the results are presented with appropriate tables and figures and clearly presented in the narrative with reference to tables and figures. Figures and tables are well labelled and appropriate for the data.

6. Discussion: Focuses on how the research questions are answered or otherwise by results organized by the 2 main study arms (purposive and cluster random sampling) and sub-arms (community microfilaria, Mini-TAS and Xenomonitoring). The authors used appropriate literature to situate the work in the context of other studies.

7. Conclusion: Is quite brief but the elaborate discussion section makes up.

Areas for authors' consideration

1. The authors elaborately describe and present advantages of the Mini-TAS (line 398-404, 426-436). It is suggested that authors could consolidate this section with description of Mini-TAS in the introduction (lines 74-78) under the introduction section.

2. The authors should clarify and ensure consistency in the decision rule for classifying a district as endemic for the purposive sampling method. Is the decision based on site prevalence (lines 420-421) or overall/cumulative prevalence in the district (lines 211-213, Table 9). Compare WHO convenience two-site method (lines 393-395).

3. The authors should reconsider use of total district population as denominator to estimate cost per capita (Lines 363-368, Table 10) which may result in underestimation and affect comparison with other studies. For both methods (purposive and cluster, including sub-arms of cluster) the entire district population was not used/enumerated/sampled. Only population of sites (villages/wards and clusters - communities/schools) were involved. For Mini-TAS and community mf survey, listing of filarial cases was not of prior relevance, hence only selected clusters (communities/schools) were used. The Xenomonitoring arm used 150 households from the 30 clusters. Cost per sample may be better estimate.

4. Lines 111, 233-238, 243-247, Tables 1, 2, 5, 6 - Authors should clarify whether filarial case data from district (NVCBDC and ASHA) were validated or collected by trained health workers to merit comparison with filarial cases collected by research team.

Other considerations

1. Lines 31 - "five top sites' would be helpful for reader at this stage to indicate ranking is based on filarial case counts.

2. Lines 32, 35 - clarify the use of "adults" in relation to sample age ≥5 years and ≥ 10 years.

3. Line 72 - reconcile 8 sites in the national protocol (purposive sampling) and 5 sites used in study.

4. Line 102 and elsewhere - consistency in use of "cluster sampling"

5. 300-301 - It would be helpful for comparison with other studies if the authors specify age group considered as adults here.

6. Standardize abbreviations NVCBDC (use in text and Tables 1 and 5)

7. Adopting INR or Rs would be helpful to readers and for consistency.

8. Number of mosquito pools in Balangir - 300 or 305 (Tabe 10, lines 339)

9. Line 385 - suggest authors use 'validation' for LF elimination as a public health problem as in earlier sections instead of certification.

10. Line 454 - since not MDA has taken place in the district the authors may reconsider "residual infection" to remove any ambiguity.

11. The reviewer has limited competency to review Lines 184-199: Filarial DNA extraction from mosquito pools.

Reviewer #3: Please see attachment file with my comments. In general, I think it is useful to make it more clear to the reader about the design of the study. In my opinion the study simply employed a cross-sectional design with two sampling strategies – purposive and random cluster sampling to evaluate the performance of four (4) sampling methods for LF remapping mapping in India. The four surveys methods are: mini-TAS, community-based household surveys of individuals aged from 5 years (national remapping protocol), community-based household surveys of individuals aged from 10 years in villages/wards with schools for mini-TAS, and entomology based molecular xenomonitoring.

6. PLOS authors have the option to publish the peer review history of their article (what does this mean?). If published, this will include your full peer review and any attached files.

Reviewer #1: No

Reviewer #2: No

Reviewer #3: No

---

## [Author Response · Author response to Decision Letter 0]

25 Sep 2023

RESPONSE TO REVIEWERS 

Reviewer #1 

Thank you for the opportunity to review the manuscript with topic; Mini-TAS as a confirmatory mapping tool for remapping areas with uncertain Filarial endemicity to exclude/ include for Mass Drug Administration: A report from field validation in India.

This is a well-written manuscript. It is a critical topic in the history of NTDs and addresses one of the critical challenges of the WHO NTD Roadmap 2030. Done in a logical manner that makes it easy to follow and comprehend. It answers an important question critical to the endgame challenges of lymphatic filariasis elimination.

This is a decision-making framework. This study compares two mapping protocols to identify more sensitive and cost-effective mapping methods. The study also took the opportunity to assess the feasibility of alternative methods involving mosquito vector and the human adult populations.

The abstract is succinct and on point and bears a lot of relevance to the NTD community. Considering the 2030 milestones and targets of the WHO NTD Roadmap makes this study time sensitive and critical to the decision making process for India and other LF endemic countries. Even more critical India has set a timeline of 2027 for elimination of lymphatic filariasis from India. It should however be noted and placed in the right perspective that WHO NTD Roadmap has estimated that 81% and not 100% of LF endemic countries would have been validated as having eliminated lymphatic filariasis as a public health problem globally.

The manuscript is strengthened by taking each of the surveys conducted in isolation and comparing the results to arrive at its conclusions and recommendations. The sampling methods applied are cost-effective and feasible to undertake under programmatic conditions. In addition, the methods described in the manuscript are adaptable to the local country settings. This strengthens the paper as a source of guidance to that process. Therefore, the processes and methods suggested in this paper provide a framework for adaptation by the WHO and programs.

The statistical analysis is well-done and easy to replicate in tandem with WHO mini-TAS guidelines. The discussion is also to the point adds value to the manuscript. It leads to valid conclusions and therefore recommendations. However, considering the objective of this manuscript, clarity should be provided on the recommendations and way-forward for these findings.

Comment & Answer 

Comment: Clarity should be provided on the recommendations and way-forward for these findings.

Answer: The recommendations and way forward are clearly mentioned as suggested

The manuscript is very useful and relevant to the current strategies of the WHO NTD 2030 Roadmap and on elimination of lymphatic filariasis as a public health problem. The conclusions, lessons learned and recommendations will be of immense assistance to the normative functions and guidelines development of the WHO for the global programme for elimination of lymphatic filariasis. The is a well-written manuscript. I recommend that it for publication after a few errors detected in the manuscript are corrected.

Reviewer #2

Areas for authors' consideration

Comment 1. The authors elaborately describe and present advantages of the Mini-TAS (line 398-404, 426-436). It is suggested that authors could consolidate this section with description of Mini-TAS in the introduction (lines 74-78) under the introduction section.

Answer: Accepted and the description of Mini-TAS is added in the introduction section 

Comment 2. The authors should clarify and ensure consistency in the decision rule for classifying a district as endemic for the purposive sampling method. Is the decision based on site prevalence (lines 420-421) or overall/cumulative prevalence in the district (lines 211-213, Table 9)? Compare WHO convenience two-site method (lines 393-395).

Answer: The decision rule for classifying the implementation unit as endemic for the purposive sampling method is that the site prevalence of Mf prevalence is above 1%. Changes are made in the line 229-231of the revised manuscript.

The results of WHO Mf prevalence in two-site method of high morbidity are compared.

Comment 3. The authors should reconsider use of total district population as denominator to estimate cost per capita (Lines 363-368, Table 10) which may result in underestimation and affect comparison with other studies. For both methods (purposive and cluster, including sub-arms of cluster) the entire district population was not used/enumerated/sampled. Only population of sites (villages/wards and clusters - communities/schools) were involved. For Mini-TAS and community mf survey, listing of filarial cases was not of prior relevance, hence only selected clusters (communities/schools) were used. The Xenomonitoring arm used 150 households from the 30 clusters. Cost per sample may be better estimate.

Answer: Cost per sample is given as suggested 

Comment 4. Lines 111, 233-238, 243-247, Tables 1, 2, 5, 6 - Authors should clarify whether filarial case data from district (NVCBDC and ASHA) were validated or collected by trained health workers to merit comparison with filarial cases collected by research team.

Answer: The filarial cases reported by the programme and ASHAs were validated by enumerating during the house hold enumeration before the start of the study (line number 125) and reported, as suggested.

Other considerations

Comment 1. Lines 31 - "five top sites' would be helpful for reader at this stage to indicate ranking is based on filarial case counts.

Answer: As suggested, “five top sites”, based on filarial case counts are added 

Comment 2. Lines 32, 35 - clarify the use of "adults" in relation to sample age ≥5 years and ≥ 10 years.

Answer: Clarified as suggested 

Comment 3. Line 72 - reconcile 8 sites in the national protocol (purposive sampling) and 5 sites used in study.

Answer: Only the national guidance, India suggest four sentinel and four random sites for mapping a district for LF. WHO initial mapping protocol suggest only two sites for Mf survey. Later, the concept of MiniTAS was suggested. For comparing two survey designs and four methods, we selected five sites from each district for the study, based on convenient sampling. 

Comment 4. Line 102 and elsewhere - consistency in use of "cluster sampling"

Answer: Two sampling strategies (purposive and random) and four methods were used and followed consistency in the use of “cluster sampling” as suggested 

Comment 5. 300-301 - It would be helpful for comparison with other studies if the authors specify age group considered as adults here.

Answer: Age group considered in cluster sampling was >10 years and referred as adults for Mf survey

Comment 6. Standardize abbreviations NVCBDC (use in text and Tables 1 and 5)

Answer: Standardize abbreviation “NCVBDC” is used in text and tables as suggested.

Comment 7. Adopting INR or Rs would be helpful to readers and for consistency.

Answer: INR was adopted for consistency as suggested.

Comment 8. Number of mosquito pools in Balangir - 300 or 305 (Tabe 10, lines 339)

Answer: Mosquito pool size was 300 and corrected in the text 

Comment 9. Line 385 - suggest authors use 'validation' for LF elimination as a public health problem as in earlier sections instead of certification.

Answer: “Validation for LF elimination as a public problem is used 

Comment 10. Line 454 - since not MDA has taken place in the district the authors may reconsider "residual infection" to remove any ambiguity.

Answer: “residual infection” word is removed and used Mf carriers 

Comment 11. The reviewer has limited competency to review Lines 184-199: Filarial DNA extraction from mosquito pools.

Answer: Only the methodology of detecting vector infection is given 

Reviewer # 3

Specific comments

Comment 1. The abstract needs to be revised for clarity. Reading the methodology and results, one notes that there were four (4) survey methods being compared as mentioned in the general comment above.

Answer: The abstract is revised. The four survey methods (purposive sanpling, MiniTAS, random sampling of human and mosquitoes), considered in the study are given clearly in the abstract. 

Comment 2. Abstract, line 32: In the methodology, line 151, the Mf survey under the mini-TAS areas was not actually among adults only, but community members 10 years old and above. 

Answer: Agreed the Mf survey in Mini TAS areas was among adults 10 years and above and corrected in the manuscript

Comment 3. Abstract, line 30-33: The statement that reads, ‘Purposive sampling … sampling’ is not clear. 

Answer: Agreed and now corrected. 

Comment 4. Abstract, line 33-36: The statement that reads, ‘Costing of … Mini-TAS’ is not clear. 

Answer: Accepted and modified in the revised manuscript 

Comment 5. Introduction, line 76: The references cited (6 and 7) do not include any WHO guidance, but rather operational research studies in Ethiopia and Tanzania in sub-Sahara Africa 

Answer: Agreed and corrected accordingly. 

Comment 6. Methodology, Study area, lines 94 and 97: Check the position of commas among the digits in the population numbers 

Answer: Accepted and revised. 

Comment 7. Methodology, Sampling strategies and Survey plans, line 106: Replace the word ‘uses’ with ‘used’ 

Answer: Replaced as suggested 

Comment 8. Methodology, Purposive sampling, line 116: In the introduction, line 72, it is stated that eight (8) sites are purposively sampled in the national remapping protocol. Why were five (5) sites sampled for this study? 

Answer: It was a research study covering two districts are for comparison only five sites from each study area. WHO suggested only two sites per district for impact assessment (preTAS).

Comment 9. Methodology, Purposive sampling, line 120: Replace the word ‘aging’ with ‘age’ 

Answer: Replaced.

Comment 10. Methodology, Purposive sampling, line 122: Revise the word ‘sapling’ to ‘sampling’ 

Answer: Replaced as suggested 

Comment 11. Methodology, Mini-TAS, line 137: The reference cited is not appropriate for Mini-TAS – it is for an LF entomology handbook 

Answer: Accepted 

Comment 12. Methodology, Mini-TAS, line 143-145: This statement reading ‘In sites … sampling’ is not clear. The sampling for mini-TAS should be done as per the Confirmatory Mapping Survey Sample Builder (SSB) tool (https://www.cor-ntd.org/resources/confirmatory-mapping-survey-builder). It is important to cite this resource if it was used 

Answer: Accepted and revised as suggested

Comment 13. Methodology, Molecular xenomonitoring, line171: Delete the word ‘by’ 

Answer: Accepted and done the correction.

Comment 14. Methodology, Molecular xenomonitoring, line173: Add the word ‘which’ after the close parenthesis 

Answer: Accepted and made the corrections.

Comment 15. Methodology, Filarial DNA extraction from mosquito pools, line188: Write the proper chemical name of magnesium chloride (MgCl2) 

Answer: Accepted and revised as suggested. 

Comment 16. Methodology, Data analysis, line 214-215: Correct the species name to ‘W. bancrofti’ 

Answer: Accepted and was corrected 

Comment 17. Ethical statement, line 220: Use past tense to read, ‘The study was …’ 

Answer: Accepted and made the corrections 

Comment 18. Results, Purposive sampling, Kalahandi district, line 246: Please provide a brief description about the cadre of staff referred to as ASHAs 

Answer: Accepted and done the corrections.

Comment 19. Results, Table 1: Please provide more information about the 3 data sources of chronic LF cases in the districts (NVBDCP, ASHAs, and Research) 

Answer: Accepted and done the revision 

Comment 20. Results, Tables 3, 4, 5, 7,8: Please consider presenting the results in simple tables and/or figures 

 Answer: Tables and Figures revised as suggested. 

Comment 21. Results, Purposive sampling, Kalahandi district, line 258: replace the word ‘is’ with ‘as’ 

Answer: Accepted and the word is replaced.

Comment 22. Results, line 284-285: There seems to be a missing word(s) in the statement reading, ‘As many ... females 

Answer: Accepted and corrected 

Comment 23. Discussion, line 388-390: Please check grammar in the sentence 

Answer: Accepted and checked the grammar 

Comment 24. Discussion, line 392-394: Check accuracy of statement 

Answer: Accepted and revised accuracy if statement 

Editors comments

 Answer: The manuscript is presented according to the PLOS ONE's style. 

2. We note that Figures 1, 3, 4, 6, and 7 in your submission contain [map/satellite] images which may be copyrighted. All PLOS content is published under the Creative Commons Attribution License (CC BY 4.0), which means that the manuscript, images, and Supporting Information files will be freely available online, and any third party is permitted to access, download, copy, distribute, and use these materials in any way, even commercially, with proper attribution. For these reasons, we cannot publish previously copyrighted maps or satellite images created using proprietary data, such as Google software (Google Maps, Street View, and Earth). For more information, see our copyright guidelines: http://journals.plos.org/plosone/s/licenses-and-copyright.

Answer: The Figure 1 titled ‘Map of Odisha state showing the location of two study districts.’ .The map of Odisha was downloaded from www.d-maps.com (https://d-maps.com/carte.php?num_car=272264&lang=en) . It is mentioned in their website that the maps are modifiable, free for any use - even commercial - under the following conditions:

-The exact URL where the original map comes from must be mentioned. Therefore, we have mentioned the exact URL on the ligands of figure 1. Additionally, as suggested, we got in touch with the original copy write holders of the map and we have received the permission to use the map in the ‘Content Permission Form’. The form is uploaded as ‘other’ file during re-submission.

For figures 3, 4, 6, 7 we have supplied replacement figures. The shape file of Kalahandi and Balangir districts of Odisha was downloaded from Balk, D., M. R. Montgomery, H. Engin, N. Lin, E. Major and B. Jones. 2020. Spatial Data from the 2011 India Census. Palisades, NY: NASA Socioeconomic Data and Applications Center (SEDAC). https://doi.org/10.7927/gya1-wp91. Accessed on Sept 11th 2023. No copyrighted material was used. The data was further modified using QGIS software version 3.4.13.

---

## [Decision Letter · Decision Letter 1]

17 Oct 2023

Mini-TAS as a confirmatory mapping tool for remapping areas with uncertain Filarial endemicity to exclude/ include for Mass Drug Administration: A report from field validation in India

PONE-D-23-20937R1

Dear Dr. Bal,

We’re pleased to inform you that your manuscript has been judged scientifically suitable for publication and will be formally accepted for publication once it meets all outstanding technical requirements.

Kind regards,

Dziedzom Komi de Souza, Ph.D.

Academic Editor

PLOS ONE

Additional Editor Comments (optional):

Reviewers' comments:

Reviewer's Responses to Questions

**Comments to the Author**

1. If the authors have adequately addressed your comments raised in a previous round of review and you feel that this manuscript is now acceptable for publication, you may indicate that here to bypass the “Comments to the Author” section, enter your conflict of interest statement in the “Confidential to Editor” section, and submit your "Accept" recommendation.

Reviewer #1: All comments have been addressed

Reviewer #2: All comments have been addressed

2. Is the manuscript technically sound, and do the data support the conclusions?

Reviewer #1: Yes

Reviewer #2: (No Response)

3. Has the statistical analysis been performed appropriately and rigorously? 

Reviewer #1: Yes

Reviewer #2: (No Response)

4. Have the authors made all data underlying the findings in their manuscript fully available?

Reviewer #1: Yes

Reviewer #2: (No Response)

5. Is the manuscript presented in an intelligible fashion and written in standard English?

Reviewer #1: Yes

Reviewer #2: (No Response)

6. Review Comments to the Author

Reviewer #1: Thank you for the opportunity to take a second look at your manuscript. I did not have many comments to address from my first review. However, I think this current version is a marked improvement on the first version with the inputs and comments from the other reviewers. The is a great improvement in the presentation and grammatical errors largely corrected. I however think the manuscript with benefit from further editorial work for publication. The abstract, as indicated earlier is well-written and to the point. The introduction provides a good background information to the topic on the Global Programme for the Elimination of Lymphatic Filariasis (GPELF), the assessment methods of LF transmission for taking treatment decisions.

the study method is adequate, based on which the analysis was done and results obtained. This aligns well with the discussions and conclusions. The recommendations and path forward is deducible from the discussion and conclusions. The study identifies mini-TAS as the most sensitive and probably cost-effective method for identifying LF infection in the end game among untreated populations.

This is a good manuscript with very useful information to direct path of the global programme to eliminate lymphatic filariasis as a public health problem.

I recommend the paper for publication without hesitation after additional editorial work.

Reviewer #2: (No Response)

7. PLOS authors have the option to publish the peer review history of their article (what does this mean?). If published, this will include your full peer review and any attached files.

Reviewer #1: No

Reviewer #2: No

---

## [Editor Report · Acceptance letter]

27 Oct 2023

PONE-D-23-20937R1 

Mini-TAS as a confirmatory mapping tool for remapping areas with uncertain Filarial endemicity to exclude/ include for Mass Drug Administration: A report from field validation in India 

Dear Dr. Bal:

I'm pleased to inform you that your manuscript has been deemed suitable for publication in PLOS ONE. Congratulations! Your manuscript is now with our production department. 

Kind regards, 

on behalf of

Associate professor Dziedzom Komi de Souza 

Academic Editor

PLOS ONE